# Not All Attention Is All You Need

## Abstract

Beyond the success story of pre-trained language models (PrLMs) in recent natural language processing, they are susceptible to over-fitting due to unusual large model size. To this end, dropout serves as a therapy. However, existing methods like random-based, knowledge-based and search-based dropout are more general but less effective onto self-attention based models, which are broadly chosen as the fundamental architecture of PrLMs. In this paper, we propose a novel dropout method named AttendOut to let self-attention empowered PrLMs capable of more robust task-specific tuning. We demonstrate that state-of-the-art models with elaborate training design may achieve much stronger results. We verify the universality of our approach on extensive natural language processing tasks.

## 1 Introduction

Self-attention network (SAN) empowered models like Transformer [1] have achieved remarkable success in recent natural language processing, which have been broadly chosen as basic architecture in a series successful pre-trained language models (PrLMs) such as BERT [2], RoBERTa [3], ALBERT [4], ELECTRA [5], DeBERTa [6] and GPT [7].

SAN has drawn a great deal of curiosity on its conceptually simple but powerful attention mechanism. However, SAN still remains a black box and more and more works attempt to unveil its inner principle, where the biggest mystery lies in its attention matrix. Our work is inspired by several recent discoveries which turn our views up and down. [8, 9] show that fixed Gaussian or even random alignment attention matrix may rival standard SAN, while more recently, [10, 11] prove that SAN may encounter a rank collapse with deepening of layers. A more concrete explanation is information diffusion [12], which states that the input vectors are progressively assimilating through continuously making self-attention. We attribute these problems to the sever co-adaption [13] between attention elements, a form of over-fitting onto SAN. As a result, self-attention empowered PrLMs hardly bring into their full play, especially for the fine-tuning stage, where task-specific data is always with limited capacity.

Dropout [13] serves as a therapy to deal with the problem, by randomly shutting down a set of units during training stage. When specified on self-attention, dropout is equivalent to adding attention mask to the attention matrix. However, random-based dropout methods like vanilla Dropout [13] or DropConnect [14] are all subject to a pre-defined distribution like Bernoulli or Gaussian, longing for exhaustive grid search for an optimal probability. Thereby a variety of works attempt to utilize manual attention mask to obtain a more informative attention matrix [15, 16], whereas all these methods require prior knowledge on model or data, which could be costly or unavailable. More recently, the rise of Neural Architecture Search [17, 18] gives birth to search-based dropout [19], which automatically chooses an optimal dropout pattern based on additional validation performances. However, the huge search space brings heavy consumption and more importantly, the obtained dropout pattern is still fixed with a pre-defined probability, which is static and sample-independent, ignoring the dynamics within different samples. In this paper, we focus on task-specific tuning of self-attention

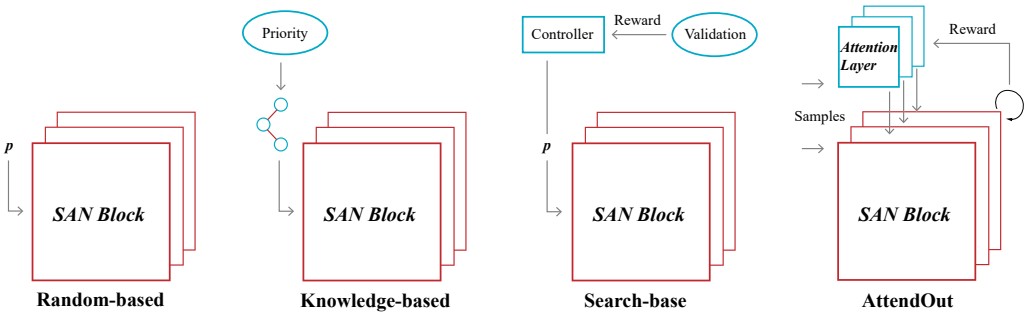

Figure 1: A diagram of different dropout methods, where $p$ refers to the dropout probabilities while $R$ refers to the reward in reinforcement learning.

empowered PrLMs and propose a novel dropout method named AttendOut onto attention layers, which leverages self-attention to dynamically generate dropout patterns for each attention layer as well as each sample through an end-to-end manner. We demonstrate that the previous state-of-the-art models with elaborate training design may achieve much stronger results. We verify the universality of our approach on extensive natural language processing tasks. Guided by AttendOut, we propose another two attention regularizers to enable simple but effective performance boost with no additional cost.

## 2 Related Work

Dropout is proposed to alleviate over-fitting problem in DNNs. Apart from vanilla Dropout [13] and DropConnect [14] which randomly shut down a subset of activations or hidden weights, there are a variety of dropout methods proposed, e.g. Alpha Dropout [20], Variational Dropout [21, 22], Adversarial Dropout [23], Energy-based Dropout [24]. However, random-based dropout encounters slow experiment cycle due to inevitable grid search. Inspired by Neural Architecture Search [17, 18], [19] proposes AutoDropout to automate the process of designing dropout patterns. A similar line of work is dynamic tuning of dropout, which further allows adaptive dropout probabilities under different training moments. [25] proposes Concrete Dropout with continuous relaxation under Concrete distribution, [26] proposes Learnable Bernoulli Dropout under discrete Bernoulli distribution using Augment-REINFORCE-Merge estimator [27], while [28] proposes Context Dropout by optimizing the evidence lower bound.

With self-attention network continuously stands out, dropout is being explored onto self-attention based models. LayerDrop [29] randomly removes entire SAN blocks, while DropHead [30], Head-Mask [31] randomly remove certain attention heads. UniDrop [32] unifies these dropout methods, which facilitates text classification and machine translation tasks. Additionally, prior knowledge is shown highly effective for guiding attention dropout as in SG-Net [15] and SIT [16], which intentionally discard syntax-unrelated attention units with the help of structural clues.

## 3 Preliminaries

In this section, we provide the preliminaries for the proposed approach. We first review the details of self-attention proposed in [1]. Based on the specific architecture, we elaborate the concerned attention dropout.

### 3.1 Self-Attention

Generally, a standard SAN block is mainly composed of an attention layer and several feed-forward layers (actually there are residual connection, layer normalization, etc. as well). The input of it is a sentence or batch of sentences of length $n$, which is first embedded through an embedding layer. The embedded input $E$ may go through three linear projections $W_Q$, $W_K$ and $W_V$ referring to query, key

and value layers respectively, and then obtain three matrices $Q$, $K$ and $V$ referring to the query, key and value components of self-attention. Subsequently, a dot-product of $Q$ and $K$ is taken and then normalized using $Softmax$ function to obtain the attention matrix $A$. Then another dot-product of $A$ and $V$ follows. The mentioned calculation can be formalized as follow:

$$Attention(Q, K, V) = Softmax\left(\frac{Q \cdot K^T}{\sqrt{d_k}}\right) \cdot V \tag{1}$$

where $\sqrt{d_k}$ is a scaling factor. Finally, the self-attention layer ends up with a linear projection $W_O$ to output.

During the aforementioned process, we highlight a key phases, that is the attention matrix $A$, which is a dot-product of $n \times n$ from two separate linear projections $W_Q$ and $W_K$. $A$ is viewed as a feature map which stores the node-to-node significance in different scores. Various works show that there hides implicit but highly needed semantic clues.

## 3.2  Dropout on Self-Attention

Our dropout will apply to the attention matrix of the concerned attention layer. We first define two specific dropouts onto Eq. 1, where both implementations are just as simple as in standard dropout via a mask matrix $M$.

**Weights Dropout.**  Weights dropout is applied to the attention matrix after $Softmax$ function by default, which is formulated as:

$$Attention(Q, K, V) = \left(Softmax\left(\frac{Q \cdot K^T}{\sqrt{d_k}}\right) \odot M\right) \cdot V \tag{2}$$

where $M$ is a binary matrix with elements in $\{0, 1\}$ and $\odot$ refers to element-wise multiplication.

**Scores Dropout.**  Different from weights dropout, scores dropout is applied before $Softmax$ function, which is formulated as:

$$Attention(Q, K, V) = Softmax\left(\frac{Q \cdot K^T}{\sqrt{d_k}} + M\right) \cdot V \tag{3}$$

Since the outer $Softmax$, we conduct an addition instead of multiplication, where elements in $M$ are set to 0 for kept units and $-inf$ for removed ones. Note that the $Softmax$ takes a similar function as the scaling factor of $1/p$ in vanilla Dropout [13], which balances the expectation of the network.

Weights dropout is commonly used in self-attention based models, while scores dropout is less explored, which is our focus in this paper. For scores dropout, we need to pay attention to a special case, when all attentions are shut down, that is, all elements in $M$ equal to $-inf$ at the same time. Such case can be formulated as follow:

$$Attention(Q, K, V) = Softmax\left(M\right) \cdot V \tag{4}$$

Note that $Softmax\left(M\right)$ obtains to a constant matrix, where each unit equals to $1/n$. In this case, the attention matrix is fixed and consequently the $W_Q$, $W_K$ and dot-product in between are skipped.

## 4  Methodology

In this paper, we propose *Attention differentiable dropOut* (AttendOut), which contributes technique novelty in the following way: (1) dynamic and task-specific tuned; (2) end-to-end trained; (3) gradient optimized dropout method onto self-attention empowered PrLMs. We elaborate our approach with two parts, in which the first is composition, while the second is training algorithm.

### 4.1  Elements of AttendOut

Our training architecture is composed of three modules, A-Net (Attacker), D-Net (Defender) and G-Net (Generator). D-Net and A-Net are two identical models and trained simultaneously through standard gradient descent, while G-Net is a learnable dropout maker and trained through policy gradient. Now we elaborate each of them.

**Defender - Attacker** As suggested, defender and attacker are two competitors playing a game with each other on specific criteria, e.g. training accuracy, training loss. Specifically, D-Net and A-Net are two identical self-attention empowered PrLMs, e.g. BERT, RoBERTa. However, they follow different dropout strategies. D-Net receives regular dropout as default in specific models, while A-Net receives additional dropout decision from G-Net onto its corresponding attention layers.

**Generator** G-Net acts as a dropout maker through generating a mask matrix for each attention layer during training stage. As aforementioned, the common dropout strategies rely on randomness, which intends to shut down the co-adaption but not powerful enough. However, our dropout maker is an agent which is able to intelligently choose and learn dropout patterns for each sample. Specifically, after training for a fixed number of steps, we conduct evaluation for both A-Net and D-Net. When A-Net obtains a higher score than D-Net, which means attacker wins the game, G-Net will be rewarded positively. When defender wins, G-Net will be punished with a negative reward. In consequence, G-Net learns appropriate dropout patterns through the game between D-Net and A-Net, while assisting A-Net to win the game. On the other hand, A-Net needs to be stronger when training under such powerful dropout, which makes it much more robust from over-fitting. Compared to search-based dropout, G-Net is triggered by the difference between two model derivatives with and without dropout, instead of the final feedback on validation set, which makes it end-to-end-possible and sample-dependent.

The design of G-Net is the most delicate part, which is also a self-attention based model with identical number of layers with D-Net and A-Net. However, we make several improvements. 1) G-Net only exports the attention scores from attention layers with no extra output layers, from which we apply Gumbel [33, 34] to sample the actions to obtain the dropout mask. 2) G-Net only makes one-head attention and share one group of parameters for all attention layers. 3) G-Net is excluded of feed-forward layers, which may obscure the impact of self-attention [11, 10].

## 4.2 Training with AttendOut

The core of training with AttendOut is to find a way to optimize G-Net, which receives signals from the difference between D-Net and A-Net. Supposing there is a list of dropout actions by G-Net:

$$a_{1:T} = \{a_1, a_2, a_3, \cdots, a_T\}$$

where $T$ refers to the number of samples, for each action $a_t$, G-Net may achieve a reward $r_t$. The optimization objective is to maximize the overall rewards of list $a_{1:T}$, denoted as $R$, that is:

$$J(\theta_G) = E_{P(a_{1:T};\theta_G)}[R]$$

where $R = \sum_{t=1}^{T} r_t$. Since $R$ is non-differentiable, we use policy gradient to update $\theta_G$ as in [17]:

$$\nabla_{\theta_G} J(\theta_G) = \sum_{t=1}^{T} E_{P(a_{1:T};\theta_G)}[\nabla_{\theta_G} \log P(a_t|a_{(t-1):1};\theta_G)r_t]$$

The above equation could be approximated as:

$$\frac{1}{m} \sum_{k=1}^{m} \sum_{t=1}^{T} \nabla_{\theta_G} \log P(a_t|a_{(t-1):1};\theta_G)r_t$$

For a model with $n$ attention layers, each dropout decision is composed of $n$ inner decisions of each layer. Additionally, each attention layer contains an attention matrix of $l \times l$, namely $l^2$ elements dropped or kept. Thus, we denote a dropout unit as $d^{ij}$, where $i$ refers to the $i^{th}$ layer while $j$ refers to the $j^{th}$ element of the attention matrix.

However, $nl^2$ dropout units bring a huge space, which makes it impossible to calculate the joint probability. To this end, we introduce the independence assumption that each dropout unit is independent with each other. Under the relaxation, we can make the following probability likelihood:

$$\log P(a_t|a_{(t-1):1}; \theta_G) = \frac{1}{nl^2} \sum_{i,j} \log P(d_t^{ij}|d_{(t-1):1}^{ij}; \theta_G)$$

150  where the summation $\sum_{i=1}^{n} \sum_{j=1}^{l^2}$ is briefly denoted as $\sum_{i,j}$.

151  Thus, the final gradient could be formalized as:

$$\nabla_{\theta_G} J(\theta_G) = \frac{1}{m} \frac{1}{nl^2} \sum_{k=1}^{m} \sum_{t=1}^{T} \sum_{i,j} \nabla_{\theta_G} \log P(d_t^{ij}|d_{(t-1):1}^{ij}; \theta_G)(r_t - b) \qquad (5)$$

152  where $b$ is a baseline function of moving average [35]. Note that we do not apply additional regular-
153  izers like L0 and L1 penalty, which impose unnecessary bias.

154  Algorithm 1 summarizes the overall procedure of training PrLMs with AttendOut. We first initialize
155  all three networks. Note that D-Net and A-Net should be kept identical at the beginning of each
156  training step. A straightforward strategy is to choose the better one to cover the other. To add
157  randomness, we sample from D-Net and A-Net based on their evaluation performances, with higher
158  probability for the better one. Then for each step, D-Net and A-Net are fed with the same mini-batch
159  data and updated via standard gradient descent, meanwhile each batch will be cached. After training
160  for $T$ steps, which we denote as a dropout step, both D-Net and A-Net are evaluated on additional
161  validation samples, which could be development set data, noisy training data or a small split of train-
162  ing data. In this paper, we simply use development set. For efficiency, we make random sampling
163  on it to retrieve $T$ samples for evaluation. Based on the evaluation scores, G-Net is rewarded with
164  $\{r_1, r_2, r_3, \cdots, r_T\}$ and updated via Eq. 5. At the end of each dropout step, the cached samples
165  will be released and D-Net and A-Net will be re-initialized.

---

**Algorithm 1** AttendOut

**Input:** Attacker $A$, Defender $D$, Generator $G$, dropout step $T$

1:  initialize $\theta_D, \theta_A, \theta_G$, where $\theta_D = \theta_A$
2:  **for** each training step **do**
3:      $\theta_D \leftarrow \theta_D'$
4:      dropout $A$ with $G$ via Eq. 3
5:      $\theta_A \leftarrow \theta_A'$
6:      **for** each $T$ steps **do**
7:          evaluate $D$ and $A$ and reward $G$
8:          $\theta_G \leftarrow \theta_G'$ via Eq. 5
9:          initialize $\theta_D, \theta_A$ for next step
10:     **end for**
11: **end for**

Figure 2: Architecture of G-Net.

---

166

**Resource Usage**  We notice that training PrLMs with AttendOut may sacrifice time and memory
168  cost. The detailed resource usage is shown in Appendix. Taking RoBERTa as an example, the
169  algorithm requires two RoBERTa models as well as a smaller self-attention based generator, which
170  is $1/3$ of RoBERTa size. Considering cached samples, roughly speaking, AttendOut requires twice
171  graphic memory as well as twice training time compared to a single model, which is a middle speed
172  line between random-based dropout and neural architecture search (Dropout [13] < AttendOut $\ll$
173  AutoDropout [19]). However, AttendOut contributes to remarkable performance gain compared to
174  other attention dropout methods.

175  **Pre-training**  Our approach is both feasible for both fine-tuning and pre-training stage of PrLMs
176  but expensive for the latter. However, we try to serve for the most delicate part of concerned issue,
177  since pre-training is generally done on large-scale data with modest training epochs, which makes it
178  less susceptible from over-fitting.

Table 1: Results (test / dev) of GLUE sub-tasks.

| Model | SST-2 Acc | MRPC F1 | QNLI Acc | MNLI-mm Acc | CoLA Mcc |
|---|---|---|---|---|---|
| BERT | 92.9 / 92.2 | 86.6 / 86.3 | 89.7 / 88.9 | 83.3 / 84.0 | 51.2 / 58.8 |
| + AtendOut | **93.6 / 93.8** | **88.1 / 87.5** | **90.2 / 91.1** | **84.2 / 84.6** | **57.4 / 60.9** |
| RoBERTa | 95.4 / 94.4 | 90.5 / 90.2 | 92.9 / 92.0 | 86.1 / 86.6 | 61.3 / 62.5 |
| + AtendOut | **96.2 / 95.1** | **91.2 / 90.9** | **93.3 / 93.0** | **87.3 / 87.8** | **63.0 / 63.8** |

Table 2: Results of IMDB, CoNLL03, PTB and SWAG respectively.

| Model | IMDB Acc | CoNLL03 F1 | PTB F1 | SWAG Acc |
|---|---|---|---|---|
| BERT | 92.2 | 94.1 | 95.4 | 81.1 |
| + AttendOut | **92.9** | **94.7** | **96.5** | **81.6** |
| RoBERTa | 93.6 | 94.5 | 96.6 | 83.8 |
| + AttendOut | **94.2** | **95.2** | **97.3** | **84.1** |

## 5 Experimental Setup

We demonstrate the universal effectiveness of AttendOut on extensive natural language processing tasks. For all mentioned tasks, we apply our method on BERT [2] and its stronger variant RoBERTa [3]. Our implementations are based on PyTorch using *transformers* [36]. For further training details, please refer to Appendix.

Our experiments include: (1) **natural language understanding:** General Language Understanding Evaluation (GLUE) benchmark [37], a collection of nine natural language understanding tasks (here we experiment on five of them, SST-2, MRPC, QNLI, MNLI-mm and CoLA; (2) **document classification:** IMDB [38], a sentiment analysis dataset where about 15% of the documents are longer than 512 word-pieces; (3) **named entity recognition:** CoNLL2003 [39]; (4) **part-of-speech tagging:** English Penn Treebank (PTB) [40]; (5) **multiple choices question answering:** SWAG [41]. We report both test and development results for GLUE sub-tasks since the large bias between them, while development results only for all the other tasks.

Note that we only adjust the dropout steps and keep all other parameters the same for strict fair comparison. For example, the parameters we use in RoBERTa are identical with what we use in training with AttendOut including both D-Net and A-Net.

## 6 Results

### 6.1 Significance Analysis

Pictorially in Table 1, RoBERTa is strong enough as it outperforms BERT by a big margin, while AttendOut empowered RoBERTa still outperforms it on all five GLUE sub-tasks. For small-scale datasets, which are more likely to over-fit, AttendOut helps unfold remarkable performance gain (**12.1% / 3.5%** over BERT on CoLA, **1.7% / 1.4%** over BERT on MRPC). However, for large-scale one like MNLI, which tends to be more stable, AttendOut still produces considerable boost, (**1.4% / 1.4%** over RoBERTa, **1.1% / 0.7%** over BERT).

Furthermore, AttendOut is shown universally effective as in Table 2. For POS Tagging, BERT and RoBERTa have achieved very strong baselines, while AttendOut empowered ones are even stronger, (**1.1%** over BERT on PTB). Similar results are seen on document classification and NER. For SWAG, however, AttendOut seems weakly effective (**0.6%** over BERT, **0.4%** over RoBERTa).

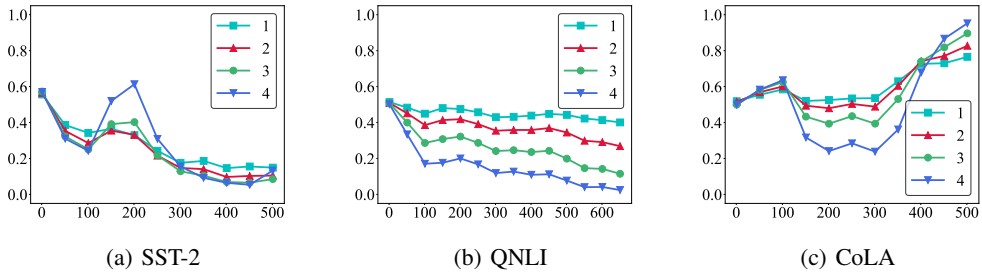

| (a) SST-2 | (b) QNLI | (c) CoLA |

Figure 3: Dropout probabilities on specific attention layers over training steps.

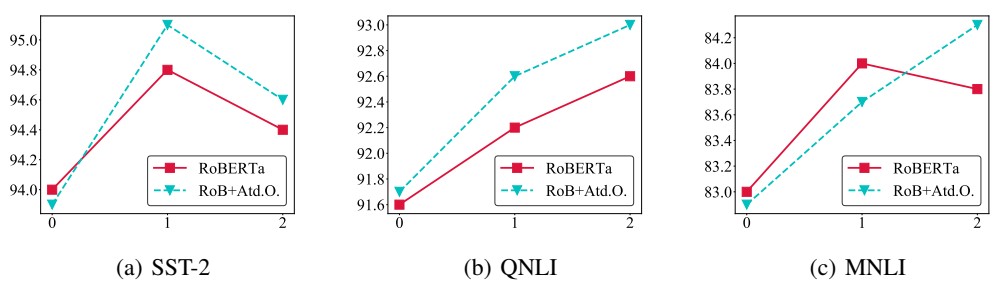

| (a) SST-2 | (b) QNLI | (c) MNLI |

Figure 4: Convergence over training epochs.

## 6.2 Visual Analysis

**Dropout Patterns** Another concerned issue is the dropout proportions by AttendOut. Figure 3 depicts the patterns on several datasets. We may find several interesting phenomenons. First, the overall patterns largely differ from datasets, which is fair since AttendOut is sample-dependent. However, we may observe something in common. Overall, the lower layers take higher dropout probabilities. For example on QNLI, the first layer almost remains steady with the probability of 0.55 during the training process, while the fourth one continuously decays in a higher rate. Intuitively, the first three layers undertake a similar trend in each dataset, while there might be an up and down for the fourth one as in SST-2 and CoLA. Especially for CoLA, we see unusual high dropout probabilities in the final period (around 0.9), which are close to complete dropout. We notice that CoLA is a small set with 8500 training samples, on which SAN model is more inclined to suffer from over-fitting. Therefore, PrLM on CoLA encounters more intensive dropout through AttendOut.

**Convergence** Figure 4 depicts the accuracy trends of RoBERTa on SST-2, QNLI, MNLI respectively. Due to a stronger dropout module, the one with AttendOut tends to fall behind (SST-2, MNLI) at the beginning of training. However, model becomes stronger since the second epoch (SST-2, QNLI). Especially on MNLI, RoBERTa obtains better results in the first two epochs and it drops in the last one, while with AttendOut, the performance is steadily rising for all three epochs.

## 7 Ablation Study

In this section, we conduct further experiments to demonstrate the effectiveness of AttendOut. Due to space limitation, we conduct corresponding experiments on development sets only.

### 7.1 Attention Dropout

**Vanilla Dropout** We conduct comparison with vanilla Dropout [13], in which we dropout the attention matrix for all layers with Bernoulli distribution of $p$. Here, we choose the dropout probabilities in $\{0.1, 0.2\}$.

Table 3: Comparison of AttendOut, vanilla Dropout and LayerDrop.

| Model | CoLA | QNLI | MNLI-mm |
|---|---|---|---|
| RoBERTa | 62.5 | 92.0 | 86.6 |
| + Vanilla | 61.3 | 92.2 | 86.9 |
| + AttendOut | **63.8** | **93.1** | **87.8** |
| + LayerDrop | 62.1 | 92.6 | 87.1 |
| + Attn.LayerDrop | **64.2** | **92.7** | **87.3** |

Table 4: Comparison of AttendOut and scheduled Bernoulli dropout.

| Model | CoLA | QNLI | SWAG |
|---|---|---|---|
| RoBERTa | 62.5 | 92.0 | 83.8 |
| + Scheduler | 63.3 | 92.6 | 83.6 |
| + AttendOut | **63.8** | **93.1** | **84.1** |

**LayerDrop** We also compare with LayerDrop [29], which focuses on skipping the entire encoder blocks, Inspired of it, we design another strategy which randomly skips attention layers via Eq. 4. For fair enough comparison, we set the dropout probabilities to 0.2 for both methods, following the settings in [29].

Intuitively in Table 3, vanilla Dropout with fixed probability does not produce noticeable gain (1.9% bellow RoBERTa on CoLA). However, AttendOut shows powerful advantage (4.1%, 1.0% and 1.0% over vanilla Dropout on CoLA, QNLI and MNLI), which stresses the necessity of dynamic dropout patterns rather than fixed static one. On the other hand, both layer-level regularizers are effective, while attention LayerDrop performs stronger and more stable on all the three. Especially on CoLA, it outperforms RoBERTa by 1.7 points, while LayerDrop meets a performance drop, which demonstrates that removing the attention layers act as a more effective regularizer than removing the entire SAN block as for self-attention based models.

### 7.2 Pattern Approximation

Guided by AttendOut, we design a dropout scheduler, in which we utilize piece-wise linearity to approximate the real curves as depicted in Figure 3. Taking QNLI as an example, we initialize the dropout probabilities to 0.6 for all attention layers and set a a specific slope for each of them. Note that here the corresponding mask matrices are randomly-generated and subject to Bernoulli distribution. In AttendOut, however, the distribution are learned dynamically through self-attention of G-Net.

As shown in Table 4, RoBERTa with scheduled Bernoulli dropout works surprisingly well on both CoLA and QNLI, which outperforms RoBERTa by 0.8 and 0.6 points respectively, closer to AttendOut, even if the strategy here is random-based and much looser. The guided scheduled dropout helps unfold the correctness of the dynamic dropout patterns learned by AttendOut as well as the self-attention based dropout maker.

## 8 Conclusion

This paper focuses on the co-adaption problem of deep self-attention networks, and presents a novel dropout method onto self-attention empowered pre-trained language models. Extensive experiments on multiple natural language processing tasks demonstrate that our proposed approach is universal and qualified to enable more robust task-specific tuning, which contributes to much stronger state-of-the-arts. We probe into the learned dropout patterns on different tasks, which empirically guide us to the very needed dynamic attention dropout design.

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
