# OpenReview forum: "Not All Attention Is All You Need"
_NeurIPS.cc/2021/Conference — NeurIPS 2021 Submitted_

### Official Review · Reviewer_Sz77 · 2021-06-30

**Rating:** 6
**Confidence:** 4

**Summary:**

The paper proposes adaptive dropout learned through reinforcement learning applied to attention in Transformer models.  Improved performance in finetuning is demonstrated on several NLP tasks.



**Ethical Concerns:**

I have no ethical concerns regarding this work.

**Limitations And Societal Impact:**

One limitation of the paper that wasn't discussed in depth is stability of the training.



**Main Review:**

Summary

The paper proposes adaptive dropout learned through reinforcement learning applied to attention in Transformer models.  Improved performance in finetuning is demonstrated on several NLP tasks.



1) Originality

I find the combination of attention dropout learned by RL novel and interesting.


2) Quality

2a) Strong points

* I find the general idea of controlling dropout through RL an interesting approach that potentially can save a lot of time when training new models.

* I like per layer dropout rate analysis from Sec 6.2.

* I also like the step made in Sec 7.2 where simpler dropout formulation is reverse engineered based on behavior of the RL policy.


2b) Weak points

* It would be nice to have confidence intervals for results in Tab 1 - 4. Finetuning usually doesn't require that much resources. I think that doing e.g. three runs should be feasible and it would greatly improve the empirical message of the paper.

* In my opinion comparing validation set accuracy obtained with AttendOut to other metrics isn't completely fair since AttendOut is using it as its reward. Therefore it is easier for AttendOut to overfit even on a valid set than for e.g. standard dropout with early stopping on valid. I still think that the method is beneficial since this argument doesn't hold for the test set. However the paper should mention this methodological issue.

* Line 192 says that dropout steps were adjusted for AttendOut. How many values were tried for each dataset? ​This leads me to the most important question. In high level this paper proposes to replace one parameter (dropout rate) by a learned adaptive module that: 1) makes training 2x slower (that might be fine) and 2) has another set of hyperparams. The question is, how much care has to be given to tuning dropout steps (the only hyperparam of AttendOut)? How stable is the training? In other words, with exactly the same compute budget would it be better to use AttendOut or some other dropout variant? I feel that the message of the paper would be greatly improved if it would be clear that for practitioners this method is clearly better than vanila dropout. At the moment this isn't the case because: A) GLUE numbers for official BERT and RoBERTa are better and it isn't obvious where the difference is comming from and B) it isn't clear how difficult it is to tune AttendOut. However both of these points can be hopefully easily adressed in the text and it might improve rating of the paper.

3) Clarity

* Line 81 --- Various works show that there hides implicit but highly needed semantic clues. --- If there are various works some citations would be nice.

* Line  141-2 --- a_{(t−1):1} should be a_{1:(t−1)} ?

* Alg 1 --- I believe that various theta' weren't defined in the text before. I assume that those are params after grad update but it isn't explicitly stated for theta_D and theta_A.

* I can't match differences from lines 200 - 202 to Tab 1. The same for line 206 and Tab 2 and for paragraph starting in line 235. Has the tables changed after writing the text?

* Fig 3 and 4: label axes of each figure.



4) Significance

The proposed adaptive dropout can be potentionaly significant for practitioners if it proves to work consistently on other datasets and it will reduce need for tuning dropout rates by hand/hyperparam optimization. In that case even ~2x slowdown of the method would be completely acceptable since hyperparam tuning often needs even more resources.


Questions:

* Why is the paper using a subset of GLUE tasks and not the full set?

* Fig 4 --- is that for dev? Why only three epochs of accuracy for QNLI and MNLI when accuracy is still improving at that point? Has it decreased in the fourth epoch?

* I am wondering whether the 0.1 and 0.2 dropout rates used in sec 7.1 for vanilla dropout are enough. E.g. Tab 3 shows that for QNLI and CoLA higher dropout rates are probably better.

* For both BERT and RoBERTa result on the official GLUE leaderboard are higher than those reported in Tab 1 (including AttendOut). Where is this difference coming from? Is it due to different implementation or was the finetuning protocol different?



Minor language issues

* Line 14 ---  a series successful pre-trained -> a series OF successful pre-trained

* Line 175 --- Our approach is both feasible for both -> approach is feasible for both

* Line 232 --- Inspired of it -> Inspired BY it?



**************************************************
Post rebutal update
**************************************************

Since many of my questions were resolved I am increasing my final score.

**Time Spent Reviewing:**

10

---

> ### Author Response · Authors · 2021-08-07
> **How difficult to tune AttendOut**
>
> Thank you for your appreciation and advice.
>
> 1. Confidence intervals
>
> Yes, we are greatly willing to report the error bars or confidence intervals in our experimental results. In the paper, we have tried five different seeds and compute the average scores in all fine-tuning tasks. Please refer to RV3 for the GLUE test results.
>
> 2. G-Net overfits on dev set
>
> Thank you for your advice. We have made experiments that using a small split of training test and dev set lead to similar results. In addition, the optimization of G-Net comes from two parts, reward and gradient. The first one comes from validation performances, while the latter one only comes from the training samples instead of validation samples. We will specify these points in our latter version. Thank you!
>
> | | SST-2 | MRPC | QNLI | CoLA |
> | - | - | - | - | - |
> | BERT | 92.9 | 86.6 | 89.7 | 51.2 |
> | ATO-dev | 93.6 | 88.1 | 90.2 | 57.4 |
> | ATO-train | 93.6 | 87.9 | 90.3 | 56.8 |
>
> 3. How much steps are tried
>
> We try dropout steps in {8, 10, 12, 15} in our early experiments. Taking SST-2 as an example, we find similar results (dev) under steps 12 and 15, while worse under 8 and 10.
>
> | T | 8 | 10 | 12 | 15 |
> | - | - | - | - | - |
> | ATO | 94.5 | 94.1 | 95.1 | 95.1 |
>
> For the influence of bigger or smaller steps, please refer to RV3.
>
> 4. How difficult to tune AttendOut
>
> It is a great question. Empirically, we suggest to train ATO with T chosen in {8, 10, 12, 15}. For the worst case, we need to fine-tune 4 times using ATO (2x computation) to obtain the best result. We roughly make this equal to normally fine-tune 8 times with different dropout rates. However, 8 times are insufficient to tune dropout rates across different layers and different samples. First, it is unpractical to choose an optimized permutation of p in different layers e.g. 0.2, 0.2, 0.1, 0.3, …. More importantly, ATO treats each attention mask as a mixed distribution of multiple Bernoulli units (l x l). Vanilla or Bernoulli dropout with fixed p or searched-based dropout cannot do such thing. That is why ATO may greatly outperform Vanilla dropout. Meanwhile the training cost is deserved.
>
> 5. Where the differences come from with AttendOut
>
> AttendOut serves like a normal dropout regularizer as well as a stronger data augmenter. We make additional experiments to show the role of ATO without other regularizers. Please refer to RV1.
>
> 6. All GLUE tasks
>
> Please refer to RV2.
>
> 7. Figure 4
>
> Yes, it is for dev, since we can not trace the accuracy of GLUE test. We keep our setting consistent with that in BERT paper, where the corresponding tasks are fine-tuned for 3 epochs (you can also refer to the Appendix).
>
> 8. Try different dropout rates
>
> We try dropout rates in {0.1, 0.2, 0.3, 0.4} in our ablation, while 0.3, 0.4 lead to very bad performances. For AttendOut, much higher rates only appear in lower layers (1 ~ 4). Besides, ATO is dynamic across different samples and layers, which makes it better even under higher dropout rates.
>
> 9. Baselines
>
> Results on GLUE leaderboard are always from large models (e.g. 24-layer RoBERT / BERT), which is why they are higher. Besides, it is a common phenomenon that the re-run baselines are different from those in original paper, because of different implementations (torch / tf) or versions. However, we use the checkpoints of popular Huggingface repo and follow the original setting in BERT paper to get our baselines results. More importantly, we use the identical hyperparameters to train the baselines as well as with ATO, which makes our experiments fair enough.
>
> 10. Clarity
>
> Citations for semantic clues
>
> Thank you for your remind!
>
> Theta’
>
> Yes, we will further clarify this part.
>
> Tables
>
> We use relative percentage in our significance analysis in Tab 1 and Tab 2. For example, 51.2 and 57.4 -> 6.2 / 51.2 = 12.1%.
>
> 11. Language issues
>
> Greatly appreciate for you careful corrections!

---

> > ### Comment · Reviewer_Sz77 · 2021-08-23
> > **Response 2 response**
> >
> > * Thank you for your response, I am increasing my score as a result of that.
> >
> > * Details about possible differences due to leaderbords results that are due to Hugging Face checkpoints should be mentioned at least in the appendix.
> >
> > * One more experiment that would fairly compare ATO and vanilla dropout would be to run hyperparam sweep with the same compute budget, e.g. ATO can have just 4 param instances while dropout would use 8 different rates with the same number of epochs. In your response to reviewer fND3 you gave each instance of dropout twice as many epochs, this would be slightly modified version of that experiment. It would again increase readers confidence that the effect is real.

---

> > > ### Author Response · Authors · 2021-08-25
> > > **Continually improve**
> > >
> > > Highly grateful for your valuable feedback and suggestions! We will continually improve our paper.

---

> > > ### Author Response · Authors · 2021-08-26
> > > **Additional experiments**
> > >
> > > Following your idea, we experiment BERT on CoLA ans SST-2 respectively. We select eight kinds of Vanilla dropout rates, comparing with four different dropout steps of AttendOut and train them within same epochs (3).
> > >
> > > The GLUE dev results are shown bellow, where p-all means the same rate across all layers. As shown in Figure 4, lower layers normally enjoy higher dropout rates. We next design three settings, 0.1 for the first and 0 for others (0.1-l1), 0.2 for the first and 0 for others (0.2-l1), 0.1 for the first 0.2 for the second and 0 for others (0.1-l1-0.2-l2).
> > >
> > > | p (Vanilla) | CoLA | SST-2 |
> > > | - | - | - |
> > > | 0-all | 59.4 | 92.3 |
> > > | 0.1-all | 58.8 | 92.1 |
> > > | 0.2-all | 58.3 | 92.3 |
> > > | 0.3-all | 57.8 | **92.4** |
> > > | 0.4-all | 57.9 | 92.3 |
> > > | 0.1-l1 | 59.4 | 92.3 |
> > > | 0.2-l1 | **61.4** | 92.1 |
> > > | 0.1-l1-0.2-l2 | 58.6 | 91.9 |
> > >
> > > | k (AttendOut) | CoLA | SST-2 |
> > > | - | - | - |
> > > | 8 | **62.0** | 92.1 |
> > > | 10 | 60.9 | **93.0** |
> > > | 12 | 59.2 | **93.0** |
> > > | 15 | 57.6 | 92.9 |

---

### Official Review · Reviewer_xozH · 2021-07-16

**Rating:** 5
**Confidence:** 4

**Summary:**

This work proposes a new adaptive dropout technique for large self-attention pre-trained language models. Their method, called AttendOut has three main elements, an Attacker, a Defender and a Generator. The Generator is trained through policy gradient to (adaptively) generate the dropout which is used in the Attacker network. The Generator network will get positive rewards when the Attacker network performs better than the Defender network in an evaluation phase. The authors test their method on a subset of tasks from GLUE, document classification, named entity recognition, POS tagging and multiple choice question answering.
Their method, which could be applied universally, achieves better results (compared to vanilla models) on the aforementioned tasks when applied on BERT and RoBERTa models.


**Limitations And Societal Impact:**

A few are discussed in the main review section.

**Main Review:**

The authors have done a good job explaining different components in their method. The algorithm explains their procedure of applying adaptive and learned dropout clearly.

The method evaluates Defender and Attacker networks every $T$ steps. One part which seems to be missing from the paper is a discussion of the relationship between $T$ and the reward that the Generator network receives. How do different values of $T$ affect how the Generator is trained?

How much training is required for the Generator to provide the Attacker with better than vanilla Dropout?

Were the rest of the tasks in GLUE tested on?

It would also be great if the authors reported error bars for different seeds to be able to realize the significance of the proposed dropout method.


**Time Spent Reviewing:**

4

---

> ### Author Response · Authors · 2021-08-07
> **Relationship between steps and rewards**
>
> Thanks for your review and advice.
>
> 1. Relationship between T steps and rewards
>
> Thank you for your questions. Smaller T results in more updates of G-Net but lower training efficiency. Contrarily, larger T encourages long-term reward in training G-Net.
>
> However, we find Ts at most 15 can just result in good performances. In the paper, we simply choose T according to the dataset size (MNLI very large -> 15, SST large -> 12, CoLA smaller -> 10) in order to guarantee convergence. Besides, we make additional experiments with different Ts on SST-2 (dev). We may specify this part in our later version. Thank you.
>
> | T | 8 | 10 | 12 | 15 |
> | - | - | - | - | - |
> | ATO | 94.5 | 94.1 | 95.1 | 95.1 |
>
> 2. How much steps required for G to provide A to outperform D
>
> It is a good question! These figures may answer your questions (drive)[https://drive.google.com/file/d/1cYVOySkABdP3sIp-Ju64woYXWMQfBZ4m/view?usp=sharing]. Pictorially, approximately 50 and 70 dropout steps for SST-2 and QNLI respectively.
>
> 3. All GLUE tasks
>
> Please see above in RV2.
>
> 4. Error bars
>
> In the paper, we have tried five different random seeds and compute the average scores in all fine-tuning tasks. We are willing to report the error bars or confidence intervals in our experimental results. Here we use standard deviation to represent the error bar.
>
> | | SST-2 | MRPC | QNLI | MNLI | CoLA |
> | - | - | - | - | - | - |
> | BERT | 92.9 | 86.6 | 89.7 | 83.3 | 51.2 |
> | ATO | 93.6 (0.2) | 88.1 (0.4) | 90.2 (0.2) | 84.2 (0.1) | 57.4 (0.6) |
> | RoBERTa | 95.4 | 90.5 | 92.9 | 86.1 | 61.3 |
> | ATO | 96.2 (0.1) | 91.2 (0.3) | 93.3 (0.3) | 87.3 (0.1) | 63.0 (0.5) |

---

> > ### Comment · Reviewer_xozH · 2021-09-03
> > **Thank you**
> >
> > Thanks for the reply and providing other results.

---

### Official Review · Reviewer_fND3 · 2021-07-16

**Rating:** 5
**Confidence:** 5

**Summary:**

This paper points out that the self-attention network can easily suffer from the co-adaption problem. To address the co-adaption problem, this paper introduces a novel dropout method, named Attention differentiable dropOut, for robust task-specific tuning. Experimental results also demonstrate the effectiveness of the proposed methods.

**Limitations And Societal Impact:**

yes.

**Main Review:**

Strengths:
1. This paper introduces an attacker-defender-generator architecture for attention differentiable dropout. The idea is interesting and experimental results verify the effectiveness of the proposed method.

====

Weaknesses:
1. The motivation of this paper is quite unclear. In the introduction, this paper first argues that self-attention remains a black box problem, and then introduces some works for self-attention by using attention masks, and some other works for linear self-attention. I cannot understand the connection between these works. And then, authors attribute these problems to the co-adaptation. However, these related works are not for addressing this problem, and the authors also do not provide the reason or relationship between these papers and co-adaption, let alone black box problem as mentioned by the authors. I think authors should re-organize the introduction to highlight the motivation.
2. Just as aforementioned, authors have attributed these problems to co-adaptation, some theoretical proof should be provided to prove the connection between co-adaption and self-attention, rather than only algorithm designing.

===

Questions:

1. Why not report all tasks of GLUE datasets? I think it is necessary to report all GLUE tasks to validate the robustness of the proposed method.
2. In my understanding, the figure 3 has shown the dropout probabilities of {1,2,3,4}-attention layer. Why not report some high layers? (For example, you can uniformly choose 1, 4, 7, 10 layer).
3. Although authors have demonstrated that the proposed method is feasible for pre-training and fine-tuning stage of pre-trained language model, some papers [2] have validated the dropout is unnecessary for large-scale pre-training and close dropout may be better for pre-training. So what is your practical value under this scenario?
4. It is unfair to compare your method with the NAS-based methods in terms of resource usage, since the searching procedure of NAS is always time-comsuming. The design of attacker and defender brings me a feeling of ensemble. Therefore, it is necessray to compare AttendOut with the vanilla dropout under the same computation.


===

Comments:
1. Line 232, "," => "."


[1] Big Bird: Transformers for Longer Sequences
[2] ALBERT: A Lite BERT for Self-supervised Learning of Language Representations

**Time Spent Reviewing:**

12

---

> ### Author Response · Authors · 2021-08-07
> **Motivation and reporting of all GLUE tasks**
>
> Thanks for your review and advice.
>
> 1. Motivation unclear and theoretical proof of co-adaption
>
> Thank you for your advice! Our work is motivated by the co-adaption problem in self-attention network. The co-adaption is early proposed in Vanilla Dropout [1], which refers to the over-dependence between neural units. The co-adaption of attention may badly affect SAN to do its job. We list some findings (line 20 ~ 23) from both empirical and theoretical angle. Based on these findings, we design our algorithm to alleviate the problem.
>
> Empirical
>
> Fixed Gaussian and random alignment attention matrix may rival standard SAN, where they actually use fixed attention weights not attention mask.
>
> Theoretical
>
> In [2], [3], authors theoretically prove that the attention output converges doubly exponentially to a rank-1 matrix with deepening of layers. In [4], author proposes information diffusion theory to explain the same thing. The attention element over-attends to each other and consequently tends to be similar in higher layers.
>
> [2], [3], [4] are consistent with our concerned attention co-adaption problem.
>
> 2. Related work
>
> We cite various dropout methods to generally address the co-adaption problem. As for attention co-adaption, there are mainly knowledge-based (SG-Net, SiT) approaches so far.
>
> Again, appreciate for your advice. We may improve our introduction to make it more organized.
>
> 3. All GLUE tasks
>
> We train all GLUE tasks in our experiments (excluding problematic WNLI as in original BERT). Due to space limitation and task diversity (GLUE are all sequence classifications), we do not report them all. The overall GLUE test results are bellow. We are willing to report them in our latter version.
>
> | |SST-2 | MRPC | QNLI | MNLI | QQP | RTE | CoLA | STS-B |
> | - | - | - | - | - | - | - | - | - |
> | BERT | 92.9 | 86.6 | 89.7 | 83.3 | 89.1 | 62.6 | 51.2 | 83.6 |
> | ATO | 93.6 | 88.1 | 90.2 | 84.2 | 89.3 | 63.5 | 57.4 | 83.8 |
> | RoBERTa | 95.4 | 90.5 | 92.9 | 86.1 | 89.5 | 71.4 | 61.3 | 89.1 |
> | ATO | 96.2 | 91.2 | 93.3 | 87.3 | 89.8 | 72.2 | 63.0 | 89.5 |
>
> 4. Dropout rates of higher layers
>
> Thank you for your advice. Our intention is that, for all datasets we find higher layers (> 6) encounter very low dropout rates (close to 0). That is why we only draw the curves of 1, 2, 3 and 4 layers, which are interesting to discuss. However, we are also willing to add higher layers into our paper for better completeness. Thank you.
>
> 5. Dropout unnecessary in pre-training
>
> We have empirically show that our dynamic dropout approach is still effective under pre-training (please refer to RV1).
>
> 6. Compare ATO and vanilla dropout under same consumption
>
> In the paper, we have compared ATO (2x computation) with vanilla dropout (1x computation). We now make additional experiments in which we fine-tune BERT with vanilla dropout for twice epochs (originally 3 epochs in our paper and BERT paper) without ATO to keep them under same computation. The test results show that much more computation does not lead to performance gain under vanilla dropout.
>
> | | SST-2 | MRPC | QNLI | CoLA |
> | - | - | - | - | - |
> | BERT (2x) | 92.9 | 86.6 | 90.2 | 51.0 |
> | ATO | **93.6** | **88.1** | **90.2** | **57.4** |
>
> 7. Ensemble
>
> Yes, we use ensemble trick to better initialize ATO, that is randomly choosing from A-Net and D-Net to initialize the weights at the beginning of each dropout steps. We set this initialization period the same as warmup proportion of lr (normally 0.06 or 0.1).
>
> 8. Line 232
>
> Thank you for your correction!
>
> [1] Dropout: A Simple Way to Prevent Neural Networks from Overfitting
> [2] Linformer: Self-Attention with Linear Complexity
> [3] Attention is not all you need: Pure attention loses rank doubly exponentially with depth
> [4] Power-bert: Accelerating BERT inference via progressive word-vector elimination

---

### Official Review · Reviewer_b59b · 2021-07-16

**Rating:** 5
**Confidence:** 4

**Summary:**

This paper proposes a novel dropout strategy for self-attention, namely AttendOut. AttendOut trains 3 independent networks.
- Network G generates a $nxn$ dropout mask for the self-attention matrix for each layer conditioned on the input data.
- Network D is trained by applying the dropout mask generated by G
- Network A is trained by applying the random dropout mask over self-attention.
Network D and A are trained with the task-specific objective. Importantly, the network G is trained with a reward defined by the advantage of D over A and optimized by the policy gradient.

Empirically, the proposed shows clear gains under language pretraining and fine-tuning.

**Limitations And Societal Impact:**

The limitation is well discussed and I don't see any concern w.r.t. negative societal impact.

**Main Review:**

The proposed idea of learning to perform input-dependent dropout is interesting. The empirical performance of AttendOut is reasonably good across a wide variety of tasks.

From the current version, there are some unclear points.
- Is the AttendOut used during pretraining or only used in fine-tuning (or vice versa)?

As acknowledged by the authors, the main concern of the proposed method is the additional computational cost.
- If it's also used for pretraining, a naive but strong baseline is to use the 2x computation consumed by AttendOut to pre-train a model without AttendOut.
- If not, the main concern would be the 2x memory issue involved in joint training may also restrict the empirical application of this method.

Another concern I have is the role of AttendOut during fine-tuning. Specifically, the G-Net may contribute to the final performance as a complement attention network (though the output is binarized). Hence, the question would be, how much does the G-Net really serve the purpose of being a regularization versus an additional network.

In addition, does the proposed model have a high variance in terms of both training stability and final performances given that it involved RL optimization?

Finally, the baseline performance reported in this work is a bit different from the original paper. For example, if we use the numbers from RoBERTa paper, the comparison will be like:

Model		MNLI	QNLI	SST-2	MRPC	CoLA

roberta.base	87.6 	92.8 	94.8 	90.2 	63.6

attention-out	87.8 	93.0 	95.1 	90.9 	63.8

Then, the gain looks very limited.

In summary, the work presents an adaptive dropout method that shows promising results on downstream tasks. There are some unclear points in the current version that clouds the judgment. Also, the method does add some computational complexity to the model training.

**Time Spent Reviewing:**

3

---

> ### Author Response · Authors · 2021-08-07
> **AttendOut is both effective for pre-training and fine-tuning**
>
> Thanks for your review and advice.
>
> 1. Use in fine-tuning
>
> Thank you for your questions. In the paper, we use AttendOut (ATO) only in fine-tuning. In this case, 2x memory cost is still acceptable. Taking GLUE for example, both fine-tuning with or without ATO can be done in only one RTX card. Meanwhile, 2x training period for tuning dropout rates is much less expensive than grid search and NAS.
>
> 2. Use in pre-training
>
> However, we recently find that ATO is still effective during pre-training stage. Based on available pre-trained checkpoint (e.g. pytorch BERT-base-uncased), we further pre-train it with ATO for one epoch, which takes 18 hours on one TiTAN RTX and then directly fine-tune it without ATO on downstream tasks. The GLUE test results are bellow (ATO-ptrn). We compare these results with further pre-training BERT for two epochs (same computation) without ATO (BERT-ptrn).
>
> |   | SST-2 | MRPC | QNLI | MNLI | CoLA | ppl (loss) |
> | - | - | - | - | - | - | - |
> | BERT | 92.9 | 86.6 | 89.7 | 83.3 | 51.2 | / |
> | BERT-ptrn | 92.9 | 87.2 | 90.2 | 83.3 | 49.9 | 5.178 |
> | ATO-ptrn | 93.3 | 88.0 | **90.3** | **84.2** | 54.1 | **5.167** |
> | ATO-fitu | **93.6** | **88.1** | 90.2 | 84.2 | **57.4** | / |
>
> 3. Role of ATO regularization
>
> This is a meaningful question. We have removed the default attention dropout when training ATO. Following your idea, we further remove L2 weight decay, and only keep the ATO and feed-forward dropout (since it has nothing to do with attention layer). The dev results are bellow (RoBERTa / ATO-nr). Due to time limitation, we apologize for not making submission for testing.
>
> | | MRPC | QNLI | CoLA |
> | - | - | - | - |
> | RoBERTa | 90.2 | 92.0 | 61.3 |
> | RoBERTa-nr | 90.0 | 91.9 | 61.1 |
> | ATO | 90.9 | 93.0 | 63.0 |
> | ATO-nr | 90.0 | 92.7 | 61.7 |
>
> Besides, we compare Attn.LayerDrop with LayerDrop and try to approximate the learned pattern in our ablation studies, which may further help explain the role of ATO regularization.
>
> Dropout regularization is proved as a weak-form data augmentation [1], while sample dependent ATO is more like the latter one and even stronger. It prevents model from over-attending to certain points, which alleviates the co-adaption.
>
> 4. High variance given RL
>
> Yes, unstable training and high variance are common in RL optimization. To this end, we introduce random sampling when initializing D-Net and A-Net (line 157). In early steps when A is not that stronger to compete with D, the initialized weights will be probably covered by D, which benefits more stable training and avoids bad high variance. We set this initialization period the same as warmup proportion of lr (normally 0.06 or 0.1).
>
> We also try 5 different seeds in our experiments. The results with error bars are shown in RV3.
>
> 5. Baselines a bit different
>
> It is a common phenomenon that the re-run GLUE baselines are different from those in original papers in recent studies. However, we use the checkpoints of popular Huggingface repo and follow the original setting in BERT paper to get our baselines results. More importantly, we use the identical hyperparameters to train the baselines as well as with ATO, which makes our experiments fair enough.
>
> [1] Dropout as data augmentation. In ICLR, 2016.

---

### Decision · Program_Chairs · 2021-09-27

**Decision:**

Reject

**Comment:**

The submission introduces an approach to preventing overfitting when fine-tuning large pre-trained NLP models. A network is trained to predict a dropout mask to improve validation performance, optimizing the network with policy gradients. Reviewers agree that the approach is interesting, and appears to be effective. However, they raised a number of concerns, including the lack of confidence intervals on results, the number of new hyperparameters introduced, only reporting results on a subset of the GLUE tasks, and several issues with the presentation of the work. I think that the author responses do promise significant improvements here, but the current submission does not meet the bar for NeurIPS.